# Real-Time Intraoperative Decision-Making in Head and Neck Tumor Surgery: A Histopathologically Grounded Hyperspectral Imaging and Deep Learning Approach

**DOI:** 10.3390/cancers17101617

**Published:** 2025-05-10

**Authors:** Ayman Bali, Saskia Wolter, Daniela Pelzel, Ulrike Weyer, Tiago Azevedo, Pietro Lio, Mussab Kouka, Katharina Geißler, Thomas Bitter, Günther Ernst, Anna Xylander, Nadja Ziller, Anna Mühlig, Ferdinand von Eggeling, Orlando Guntinas-Lichius, David Pertzborn

**Affiliations:** 1Department of Otorhinolaryngology, Jena University Hospital, 07747 Jena, Germany; ayman.bali@med.uni-jena.de (A.B.); saskia.wolter@med.uni-jena.de (S.W.); daniela.pelzel@med.uni-jena.de (D.P.); ulrike.weyer@med.uni-jena.de (U.W.); mussab.kouka@med.uni-jena.de (M.K.); katharina.geissler@med.uni-jena.de (K.G.); thomas.bitter@med.uni-jena.de (T.B.); guenther.ernst@med.uni-jena.de (G.E.); nadja.ziller@med.uni-jena.de (N.Z.); anna.muehlig@med.uni-jena.de (A.M.); ferdinand.von_eggeling@med.uni-jena.de (F.v.E.); orlando.guntinas@med.uni-jena.de (O.G.-L.); 2Department of Computer Science and Technology, University of Cambridge, Cambridge CB3 0FD, UK; tiago.azevedo@cst.cam.ac.uk (T.A.); pl219@cam.ac.uk (P.L.); 3Department of Pathology, Jena University Hospital, 453003 Jena, Germany; anna.xylander@med.uni-jena.de; 4Comprehensive Cancer Center Central Germany, 07747 Jena, Germany

**Keywords:** hyperspectral imaging, deep learning, histopathological ground truth, 3D modeling, intraoperative decision-making, tumor margin assessment, head and neck cancer

## Abstract

Our study describes a novel approach that combines hyperspectral imaging and deep learning, validated with histopathological ground truth, to enable rapid and accurate delineation of tumor margins in head and neck cancer surgeries. We present a system that acquires spectral data from freshly resected tumor samples and reconstructs three-dimensional models, which are then analyzed by convolutional neural networks. This label-free technique offers real-time tissue classification that can potentially reduce surgical times and enhance patient outcomes. By providing a precise and efficient alternative to traditional histopathology, our work opens new possibilities for more data-driven surgical decisions and improved cancer diagnostics.

## 1. Introduction

In oncologic surgery, complete tumor resection is crucial for patient prognosis, as seen in head and neck cancer. Therefore, the accurate tumor margin delineation is essential for the complete resection but cannot be reliably achieved by the surgeon alone without supplementary methods [1]. The current standard, intraoperative frozen section diagnostics, is both time-consuming and limited in accuracy, achieving 63.1% and requiring approximately 30 min per assessment [2]. Moreover, its application depends on the availability of a pathologist. During this procedure, excised tissue undergoes expedited histopathological evaluation while the patient remains under anesthesia, often leading to expensive surgical pauses [3].

No current solution adequately addresses the dual challenges of accuracy and efficiency in intraoperative tumor margin detection. Various approaches, including fluorescence imaging, microscopy, ultrasound, radiography, optical coherence tomography, magnetic resonance imaging, elastic scattering spectroscopy, bio-impedance, computed tomography, mass spectrometry, Raman spectroscopy, nuclear medicine imaging, terahertz imaging, photoacoustic imaging, and pH measurement, have been explored. However, no single technique meets the real-time and reliability requirements of surgical procedures, as has been explored in recent studies and reviews [4,5,6].

Among the various imaging modalities investigated for intraoperative tumor margin assessment, hyperspectral imaging (HSI) has emerged as a particularly promising technology due to its ability to capture spectral information beyond the capabilities of conventional imaging. HSI records spectral data across more than 30 wavelengths, forming a three-dimensional dataset, which enables precise distinction of biological tissues based on their unique spectral signatures. Its capacity to provide label-free, non-invasive, and non-ionizing tissue characterization makes it particularly suited for oncological applications. By leveraging near-infrared light, HSI penetrates deeper tissue layers, allowing for enhanced contrast between malignant and healthy tissues [7,8]. Tumor cells display unique spectral properties, acting as distinct identifiers for malignancy [9]. This unique capability positions HSI as a compelling alternative to conventional intraoperative assessment techniques, with the potential to improve surgical precision and patient outcomes.

In vivo HSI applications include skin cancer diagnostics by analyzing melanin distribution [10], functional imaging of blood flow and oxygen saturation [11], and skin transplant assessment [12]. Hyperspectral endoscopy further expands these capabilities using scanning [13] and snapshot methods [14].

HSI has shown potential for in vivo tumor margin delineation in image-guided surgery, with a reported sensitivity of 94%, specificity of 68%, and an AUC of 92% [15]. However, deeper tissue margins remain difficult to assess, with adequate detection in only 13% of 301 oral and oropharyngeal carcinoma resections [16].

Ex vivo biopsy analysis may offer a solution, providing high-resolution imaging without intraoperative constraints. Ex vivo HSI measurements also benefit from the stability of biopsy samples, avoiding problems like patient movement and image misregistration, allowing for mesoscopic and microscopic imaging with high spatial and spectral resolution. Correlations between HSI data and histopathological findings have been demonstrated [8]. HSI-assisted tumor surgery holds the potential of speeding up diagnostics and enhancing treatment efficiency by reducing reliance on conventional histopathology. Applicability has been demonstrated on breast [17], liver [18], brain [19], kidney [20], stomach [21], head and neck [22], and thyroid tissues [23]. With HSI biopsy samples can be analyzed within minutes, as demonstrated in breast cancer detection with 84% accuracy [24]. However, spatial orientation remains a challenge, limiting tumor margin assessment to isolated sections.

HSI generates vast datacubes, necessitating automated analysis. Both traditional feature extraction techniques and advanced artificial intelligence methods have been employed. Traditional methods—such as principal component analysis, spectral unmixing, spectral angle mapping, and other statistical techniques—have been used to extract features from hyperspectral datasets [25,26]. However, deep learning approaches, particularly Convolutional Neural Networks (CNNs), have emerged as a dominant method for processing hyperspectral data, driving significant advances in HSI-based pathology classification, notably in head and neck cancer [9,14,15,27,28].

While HSI appears to be a promising method in both in vivo and ex vivo applications, challenges persist in correlating excised tissue with its original anatomical orientation. Additionally, current approaches provide limited insight into deeper tissue layers. To address these issues, 3D HSI modelling has been proposed. However, its clinical feasibility remains limited. Yoon et al. highlight that achieving precise 3D HSI in clinical settings is a complex task, particularly in defining lesion boundaries and identifying metastasis [5]. Slicing the sample into thin sections for 3D HSI reconstruction is too slow for real-time application. Alternative methods such as spectral optical coherence tomography (OCT) [29] and multispectral photoacoustic tomography (PAT) [30] require further technological advancements [5].

In this study, multi-angle HSI was performed on fresh frozen ex vivo samples from patients with head and neck squamous cell carcinoma (HNSCC), which are prone to cancer recurrence or secondary primary tumors [31,32]. HSI images were acquired from multiple orientations to capture spectral variations across different tissue surfaces, ensuring comprehensive spectral characterization. A key strength of our approach is the integration of HSI data with high-resolution 3D histological models derived from serially annotated sections, establishing a robust ground truth for spectral classification. This enabled precise spatial correlation between HSI features and tumor boundaries and facilitated CNN training for voxel-wise tumor delineation.

## 2. Materials and Methods

The entire workflow—from sample preparation for multi-angle HSI to 3D histological reconstruction—is summarized in Figure 1, which illustrates how freshly resected HNSCC samples are transformed into a structured, dual-modality dataset for hyperspectral analysis and deep learning. To achieve this, two complementary 3D models were generated: an RGB-3D model derived from the HSI data, reconstructed from multi-angle HSI acquisitions, which captures the spectral characteristics of the tissue, and a histological 3D model, created from serially sectioned and pathologist-annotated histological slices, serving as a spatially precise ground truth for tumor classification. These models were co-registered to establish a direct voxel-wise correlation between HSI-derived spectral features and histopathological validation, forming the foundation for deep learning-based segmentation.

### 2.1. Sample Preparation for HSI 3D Modelling

Tissue samples ranging from 1 to 3 cm in width, height, and length were obtained from six HNSCC patients (Table 1) undergoing tumor surgery. Immediately after excision, each sample underwent snap freezing in liquid nitrogen to preserve tissue integrity and biochemical composition. Following this, the specimens were stored at −80 °C until further processing. For imaging and sectioning the samples were transferred to a controlled environment where they were maintained at 5 °C using dry ice to ensure minimal degradation during handling.

HSI of the samples was performed using the CE-marked TIVITA™ Mini Imaging System (Diaspective Vision GmbH, Am Salzhaff, Germany), a Class I medical device compliant with Medical Device Regulation (MDR) standards. The camera captures images at a spatial resolution of 540 × 720 pixels, spanning a spectral range of 500–1000 nm. This range is divided into 100 channels, resulting in a spectral resolution of 5 nm. Illumination was provided by integrated LEDs in the HSI system, and each image required approximately five seconds of acquisition time.

To capture the full range of spectral variations across different tissue structures, the samples were imaged from multiple perspectives. Specifically, the HSI camera captured between 30 and 40 images of each sample at top, middle, and bottom perspectives, rotating by approximately 10° increments. Each acquisition yielded a hyperspectral datacube, from which a red, green, and blue (RGB) image was computed. A corresponding RGB 3D model (Figure 1E) was then generated using photogrammetry software—such as Apple’s Object Capture API (Apple Inc., Cupertino, CA, USA) or, alternatively, AliceVision Meshroom v.2023.3.0, an open-source 3D reconstruction pipeline—that employs advanced photogrammetry techniques to reconstruct 3D objects from a series of RGB images [33]. Finally, these models were integrated into an augmented reality (AR) platform, enabling interaction with a real-time, spatially contextualized representation of the tumor, thereby enhancing both the visualization and clinical interpretation of the sample.

The one-to-one mapping between the RGB images and the hyperspectral datacubes ensured that every pixel in the RGB image was paired with its complete spectral signature from the hyperspectral dataset. For each sample, approximately 100 hyperspectral datacubes were acquired, capturing spatially resolved spectral information at various tissue depths and orientations. In total, 712 datacubes were collected from all six specimens. This dataset was used to train deep learning algorithms for spectral classification and tumor boundary detection. Each datacube provided a detailed spectral signature for every pixel, allowing voxel-wise spectral analysis for tissue characterization.

### 2.2. Sample Preparation for 3D Histology Modelling

After overnight fixation in 4% formalin, formalin-fixed paraffin-embedded (FFPE)-sectioning (Figure 1B) was performed along the Z-axis, with each section cut at a uniform thickness of 10 µm. Each sample was sectioned into 250–300 slices, resulting in roughly 1650 histological sections across all six samples. The sections were then stained using standard hematoxylin and eosin (H&E) protocols and digitally scanned at 40× magnification using the NanoZoomer System brightfield microscope scanner (Shizuoka Prefecture, Japan).

For annotation, pathologists manually labeled 10 to 15 sections per specimen to create reference data. These expert annotations served as ground truth for an automated annotation system implemented via the IKOSA Software v 5.1.1 (KOLAIDO GmbH, Thal, Switzerland), which extended the annotations to the remaining sections. The extended annotations were then manually checked for correctness.

The annotated sections were then co-registered using a custom Python script that applied affine registration. After alignment, the sections were stacked to generate a volumetric reconstruction of the sample (Figure 1F) using 3D Slicer, an open-source platform for medical image processing and visualization [34]. This reconstruction enabled a direct comparison between histological features and HSI-derived signatures. For each datacube, a 2D annotation mask was generated from the ground truth 3D model, containing three class labels (Figure 2A)—“tumor”, “healthy”, and “background”. This mask could then serve as a label for single-label classification in the following deep learning applications.

### 2.3. Deep Learning and CNN

For the task of automated classification based on the HSI data, three distinct CNN-based architectures—U-Net, U-within-U-Net (UwU-Net), and U-Net Transformer (Figure 2C)—were employed. These models were specifically chosen because U-Net-based architectures have proven to be highly effective for precise image segmentation and classification, particularly in the medical imaging domain [35]. Their ability to work well with limited training data makes them especially suitable for the present task. The goal was to classify three distinct classes: tumor, healthy, and background, in a multiclass classification problem, where each pixel in the HSI data is assigned one of these three categories. The HSI data was pre-processed by standard normal variate normalizing and converted into suitable input formats for the models. To improve the generalizability of the models, various augmentation measures were randomly applied during training, including rotation and mirroring, brightness and contrast variation, as well as random noise.

The first model, U-Net, was introduced by Ronneberger et al. and is a specialized CNN architecture designed for image segmentation. U-Net follows the principle of an autoencoder, consisting of a contracting path (encoder) and a symmetric expanding path (decoder), giving it its characteristic U-shaped structure. The contracting path is composed of multiple convolutional layers and max-pooling operations, generating feature maps while reducing spatial resolution through downsampling. The expanding path restores spatial resolution by upsampling feature maps using transposed convolutions (up-convolutions). The final output consists of feature maps corresponding to the segmented classes. Additionally, skip connections link corresponding layers in the contracting and expanding paths, enabling the combination of deep, contextual information with high-resolution details [36].

The second model, U-within-U-Net (UwU-Net), introduced by Manifold et al., builds upon the original U-Net architecture. However, it is specifically adapted for data with multiple spectral channels, such as HSI, which incorporate an additional third tensor dimension. In UwU-Net, spectral channel information is processed within a separate, outer U-shaped framework, which ensures efficient capture of spectral dependencies. By doing so, UwU-Net can accurately process the complex spectral relationships inherent in HSI data, while still preserving the spatial resolution crucial for precise image segmentation [37]. The U-Net Transformer, proposed by Petit et al., builds upon the classic U-Net architecture by incorporating self- and cross-attention mechanisms from Transformer models. The U-Net Transformer addresses limitations of traditional U-Net models, which struggle to model long-range contextual interactions and spatial dependencies. The Transformer-based attention mechanisms are integrated at two key points: self-attention operates on encoder features to model global interactions, while cross-attention in the skip connections enhances fine spatial recovery in the decoder by filtering out non-semantic features [38].

To train these models on the custom HSI dataset, a k-fold cross-validation approach was employed to account for the limited data availability. The dataset was divided into 5 consecutive folds, with each fold split using 80% of the imaging perspectives for training and 20% for testing (Figure 2B), ensuring a robust evaluation across different data subsets. Each model was trained for 150 epochs, and the version with the lowest validation loss was saved for each fold, ensuring the best-performing model was retained.

The training process utilized the cross-entropy loss function, commonly used for multi-class classification tasks. The learning rate was set to 1 × 10^−3^, and the Adam optimizer was used to update the model parameters throughout training.

### 2.4. Statistical Metrics for Evaluating Model Performance

To evaluate the model’s performance in the semantic segmentation task, several well-established metrics were computed. The evaluation was carried out on a per-pixel basis, treating each pixel as an independent classification instance. The overall accuracy is calculated as the proportion of correct predictions relative to all predictions. It provides an overall view of the model’s performance but weighs all classes equally. As a result, it can be misleading in cases of imbalanced class distributions or when the detection of a specific class is more critical.$$Accuracy=\frac{correct\;predictions}{all\;predictions}$$

To gain deeper insights, additional class-wise metrics were computed following a one-vs-rest scheme, in which the classification performance is assessed individually for each class. For every class, each pixel is categorized into one of four groups: true positives (TP)—correctly predicted pixels of the class; false positives (FP)—pixels incorrectly assigned to the class; false negatives (FN)—pixels belonging to the class but not identified as such; and true negatives (TN)—pixels correctly identified as not belonging to the class.

The class-wise recall, also referred to as sensitivity, quantifies the ability of the model to identify all instances of a class. It is defined as the proportion of TP predictions relative to the total number of ground truth pixels of that class. This metric is crucial when it is important to identify as many instances of a class as possible.$$Recall=\frac{TP}{TP+FN}$$

The class-wise precision is calculated as the proportion of TP predictions for a class relative to the total number of instances predicted as belonging to that class. This metric provides insight into the reliability of the predictions and helps assess how many of the predicted regions truly correspond to the target class.$$Precision=\frac{TP}{TP+FP}$$

To balance the trade-off between precision and recall, the F1-score was calculated as their harmonic mean. The F1-score is particularly useful when both FP and FN have significant implications, as is often the case in clinical applications. In the context of image segmentation, the F1-score is mathematically equivalent to the Dice coefficient, a widely used metric in medical imaging.$$F1-score=\frac{2\cdot Precision\;\cdot \;Recall}{Precision+Recall}=\frac{2TP}{2TP+FP+FN}$$

Concluding, all evaluation metrics were computed individually for each fold of the k-fold cross-validation procedure. The final reported values represent the mean and standard deviation across all folds, thereby reflecting both average performance and variability.

## 3. Results

### 3.1. Performance Metrics of the Deep Learning Model

Three different deep learning models, based on the U-Net architecture (classic U-Net, UwU-Net, U-Net Transformer), were trained for the automated segmentation and classification of the tissue samples. To evaluate the model performance and compare the model architectures, different metrics were calculated for each model using the test data (Table 2).

All models demonstrated the ability to classify the data reliably, achieving an accuracy above 0.90. The classical U-Net achieved the highest accuracy at nearly 0.98, followed by the U-Net Transformer with 0.96 and the UwU-Net with 0.93. The class-wise recall, which measures how well a model identifies all instances of a specific class, was also highest for the classical U-Net, reaching approximately 0.93 for tumor regions. The UwU-Net and U-Net Transformer performed slightly worse in this regard, both achieving a recall of about 0.91 for the tumor class. While these values indicate a high detection rate, the slightly lower recall suggests that these models may miss more tumor areas compared to the classical U-Net.

Class-wise precision, which evaluates the ability to minimize false positives, was particularly important for the healthy class to ensure that tumor regions were not mistakenly classified as healthy tissue. The classical U-Net achieved the highest precision for this class at around 0.75, whereas the U-Net Transformer reached 0.70, and the UwU-Net performed noticeably worse with a precision of 0.42. This suggests that the UwU-Net had more difficulty distinguishing between healthy and tumor tissue.

The F1-score, which balances recall and precision, further highlights these performance differences. For the tumor class, the classical U-Net achieved the highest F1-score of 0.90, followed by the U-Net Transformer with 0.89 and the UwU-Net with 0.84. These results indicate that while all three models successfully classified tumor regions, the classical U-Net consistently outperformed the others across all key metrics, making it the most reliable choice for this application.

### 3.2. Proof of Concept 3D Hyperspectral Imaging System

To demonstrate the feasibility of a rapid HSI biopsy assessment system (Figure 3A), we present a schematic sketch outlining the core components of the proposed setup (patent pending). Although the system’s design has not yet been finalized, we provide a proof of concept based on currently commercially available hardware validating its potential effectiveness. The proposed system comprises an HSI camera, a cooling chamber, a rotating motorized stage for positioning the biopsy, and a deep learning model trained to classify tumor presence based on hyperspectral data. The workflow is designed to minimize intraoperative delay, with a total scanning and analysis time of around 10 min. The average processing time per sample was calculated as follows: acquiring 75 HSI measurements of the sample from top, middle, and lower perspectives at 5 s each (75 × 5 s = 375 s); generating binary masks via the pretrained neural network took approximately 120 s; and reconstructing the 3D tumor model from the HSI measurements using the binary masks required another ≈120 s. In total, the mean analysis time per specimen was 375 s + 120 s + 120 s = 615 s (≈10.3 min).

Upon biopsy excision, the tissue is snap-frozen to preserve molecular integrity and then placed within the cooling chamber of the system. The biopsy is incrementally rotated in 10-degree steps, enabling the HSI camera to capture spectral information from multiple angles, including top, middle, and bottom views, providing a full 360-degree assessment. The hyperspectral data is processed through a CNN, which generates a real-time tumor segmentation and classification map. This enables the surgeon to receive a binary assessment—whether residual tumor cells remain in the biopsy or if the sample is tumor-free—within a clinically feasible timeframe.

To validate this concept, we applied our approach to samples from patients diagnosed with HNSCC of the oropharynx, oral cavity, hypopharynx, and nasal cavity. The results (Figure 3C) demonstrate that the trained CNN model can effectively delineate tumor regions within the sample, supporting the feasibility of an intraoperative decision-support system that can be installed directly in the operating room to facilitate immediate decision-making, or alternatively, positioned in an adjacent location to ensure rapid diagnostic feedback. This was achieved by rendering the resected tissue as a spatially contextualized 3D model (Figure 4). Here, we present findings from an AR-integrated approach, demonstrating its potential to refine surgical decision-making and improve spatial understanding of tumor margins.

## 4. Discussion

While recent studies have advanced the application of hyperspectral imaging (HSI) in head and neck cancer surgery—for example, demonstrating feasibility in intraoperative diagnostics with analyses of tongue cancer cases [26,39,40] and achieving 81% accuracy in deep learning-based classification of head and neck cancer patients [41]—our study further explores this potential by integrating accurate histopathological ground truth data. The results of our work highlight that combining HSI with deep learning enables real-time intraoperative tumor margin assessment. Although our system is currently at a proof-of-concept stage, it demonstrates that HSI data can be used effectively to classify tumor presence with high accuracy, offering a non-invasive, label-free alternative to conventional histopathology. The fast throughput time (<10 min) and intraoperative deployment ensure minimal disruption to surgical workflow. Moreover, our study successfully classified HNSCC across multiple anatomical regions using deep learning models (U-Net, UwU-Net, and U-Net Transformer), underscoring the flexibility of this approach. Future work will focus on optimizing the deep learning framework, expanding the dataset to include more tumor subtypes, and transitioning from a simulated workflow to a fully integrated clinical system.

Comparable studies have demonstrated that HSI can detect head and neck cancer [8,39,42] and HSI is gaining considerable attention as a medical imaging modality due to the increasing availability of systems, combined with lower costs and complexity associated with them [5]. However, many earlier investigations were hindered by prolonged measurement times, and complex computational methods [43]. Moreover, research employing similar imaging platforms has predominantly concentrated on perfusion parameters, thereby limiting broader diagnostic applicability [44]. Beyond these workflow advantages, HSI may offer enhanced detection accuracy compared to autofluorescence or fluorescence imaging using agents such as 2-NBDG and proflavine [45]. Alternative optical modalities have also been explored. For instance, narrow-band imaging (NBI) leverages specific spectral bands (400–430 nm and 525–555 nm) to accentuate vascular structures [46,47]. Although a systematic review found insufficient evidence to support NBI’s superiority over white-light-guided surgery in determining safe mucosal margins, its targeted spectral approach may inform the refinement of hyperspectral protocols for faster imaging [46]. Similarly, confocal laser endomicroscopy (CLE) offers high-resolution, cellular-level visualization with or without contrast agents [48]; however, its limited field of view (<1 mm²) restricts its clinical impact despite high accuracy [49]. In contrast, our work establishes the foundation for a novel HSI-based intraoperative diagnostic system that offers rapid, non-invasive, and label-free tumor margin assessment. By integrating real-time spectral imaging with deep learning, this approach holds significant promise for enhancing surgical decision-making and improving patient outcomes in oncologic procedures.

A key advantage of the proposed system is its ability to provide rapid intraoperative feedback, eliminating the need for time-intensive histopathological evaluation. Traditional frozen section analysis requires multiple processing steps—including snap freezing, staining, and microscopic examination—which can significantly delay surgical decision-making. In contrast, spectral analysis enabled by this system allows surgeons to make near real-time, data-driven decisions within minutes. Additionally, the use of multi-angle hyperspectral imaging ensures comprehensive biopsy surface mapping, minimizing the risk of unlabeled tumor regions and enhancing classification robustness

The integration of multi-angle HSI into a 3D tumor model is a major innovation of this study. The 3D reconstruction itself does not directly enhance classification accuracy but serves as a crucial tool for geometric consistency, aligning spectral information with histopathological ground truth.

More importantly, the 3D reconstruction pipeline enhances surgical interpretability by generating a spatially coherent tumor map that maintains the biopsy’s anatomical orientation. This addresses a key limitation of traditional 2D HSI measurements, ensuring that tumor margins are evaluated within their original anatomical context, thereby improving surgeon confidence and decision-making. Ground truth for the development of the deep learning algorithm was based on high-resolution histopathology data. This makes the development even more complex but is needed to reach sufficient accuracy for a clinical application. An annotation of HSI images based only on the surgeons’ assessment like, e.g., in [50] is not sufficient for this purpose.

One of the key challenges in this study was the limited number of available samples, which can be a significant constraint for deep learning models, as they typically require large datasets for robust training, which is a potential reason for the larger models’—UwU-Net and U-Net Transformer—poorer performance. To address this issue, k-fold cross-validation was applied, ensuring that all available data was effectively utilized for both training and evaluation. Despite this, increasing the dataset size remains a crucial step for future studies, as a larger sample pool would enhance model generalizability, reduce the risk of overfitting, and allow for more statistically robust conclusions.

Another potential strategy to improve model performance is transfer learning, where pre-training on other HSI datasets could provide a better initial weight distribution and reduce the dependency on large amounts of annotated data. This approach has shown promise in other medical imaging applications and could further optimize feature extraction for hyperspectral data [51].

To further enhance segmentation performance, future research may explore alternative architectures—such as Graph Neural Network (GNN) or various hybrid models [50,52]—to complement current approaches. While CNN-based segmentation facilitates rapid classification, it may still exhibit boundary ambiguities in regions with subtle tumor margins, suggesting a need for hybrid artificial intelligence strategies that integrate spectral unmixing with probabilistic modeling for more precise delineation. In addition, future studies should investigate whether snap freezing is essential or if simpler cooling methods could adequately preserve tissue quality, potentially streamlining the sample preparation process.

Another potential limitation is the requirement for strict imaging conditions during hyperspectral data acquisition. Inconsistency in lighting, tissue orientation, and sample freezing consistency could introduce variables that affect model performance. Standardizing these factors in a clinical workflow will be essential for ensuring consistent and reliable results.

Future work should address both the technical robustness and clinical integration of the system. Prospective clinical trials are essential to assess its stability, reproducibility, and predictive accuracy under real-world operating conditions. Equally important is the translation of these measurement results into the intraoperative setting. In surgical scenarios where further resection may be indicated based on imaging outcomes, a reliable method is required to accurately correlate the segmented tumor regions with the patient’s anatomy. One promising approach involves integrating the three-dimensional HSI data directly into the surgical field—potentially through AR (Figure 4)—to enable a real-time overlay of the 3D model and facilitate precise identification of tumor margins during surgery [53].

## 5. Conclusions

Our study shows that integrating HSI, deep learning, and 3D modeling can rapidly and accurately delineate tumor margins in cancer surgeries. This approach offers a non-invasive, label-free alternative to conventional histopathology, providing near real-time feedback (in under 10 min) to enhance intraoperative decision-making. With further refinement and clinical integration, this platform has the potential to streamline surgical workflows and improve patient outcomes.

## 6. Patents

The concept of the proposed system is the subject of a pending patent (Patent: Verfahren und Vorrichtung zur Zuordnung mindestens eines tumorrelevanten Segmentierungswertes zu jeweils einem Oberflächenpunkt einer Biopsie-Probe anhand hyperspektraler Intensitätswerte, application number: P 17403 DE) by Ayman Bali and David Pertzborn.

## Figures and Tables

**Figure 1 cancers-17-01617-f001:**
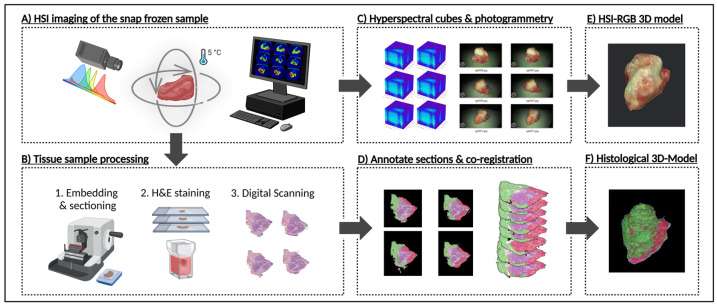
Technical workflow from frozen sample to HSI-based and histological 3D models. Workflow involves two primary steps: HSI-based 3D reconstruction, and histopathological sectioning and annotating. (**A**) HNSCC sample is first imaged using the CE-marked TIVITA™ Mini hyperspectral imaging system. Approximately 100 hyperspectral image datacubes are acquired per sample, each at ~10° rotational increments at top, middle, and bottom perspectives. (**B**) Samples are then formalin-fixed, paraffin-embedded (FFPE), sectioned at a thickness of 10 µm, stained with hematoxylin and eosin (H&E), and scanned using a 40× digital microscope scanner. (**C**) From the hyperspectral data, red, green, and blue (RGB) images are generated for photogrammetric surface modeling. (**D**) Selected histological sections are manually annotated by pathologists, while remaining sections are annotated based on pathologist’s annotations using IKOSA Software v 5.1.1. All annotated sections are co-registered through an affine registration algorithm implemented in a custom Python script. (**E**) Hyperspectral-RGB data is used to produce a high-resolution 3D surface model via Apple’s Object Capture API. (**F**) The co-registered annotated sections are reconstructed into a volumetric annotated histological 3D model. Created in BioRender Release: 24 April 2025 (BioRender Inc., Toronto, ON, Canada). https://BioRender.com/w36v028.

**Figure 2 cancers-17-01617-f002:**
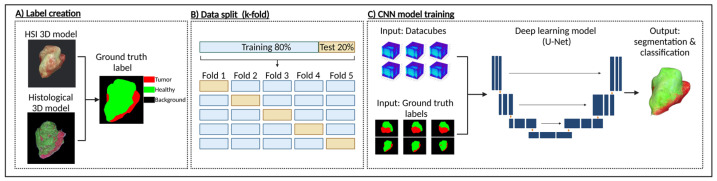
Data processing pipeline for tumor classification using HSI and deep learning. (**A**) Multi-class label creation was performed by comparing the HSI-derived RGB 3D model with the ground-truth histological 3D model. Tumor regions were identified based on histological annotations and mapped onto the hyperspectral 3D model, generating voxel-wise labels where red represents tumor tissue, green represents healthy tissue, and black marks the background. (**B**) Labeled dataset was split into training and test sets using k-fold cross-validation to ensure robust model evaluation. (**C**) Three CNN-based architectures were trained using HSI datacubes and corresponding ground truth labels as input. Trained deep learning models produced a segmentation map of the sample, where predicted tumor regions are highlighted in red and non-tumor regions in green. Created in BioRender Release: 24 April 2025 (BioRender Inc., Toronto, ON, Canada). https://BioRender.com/d19m652.

**Figure 3 cancers-17-01617-f003:**
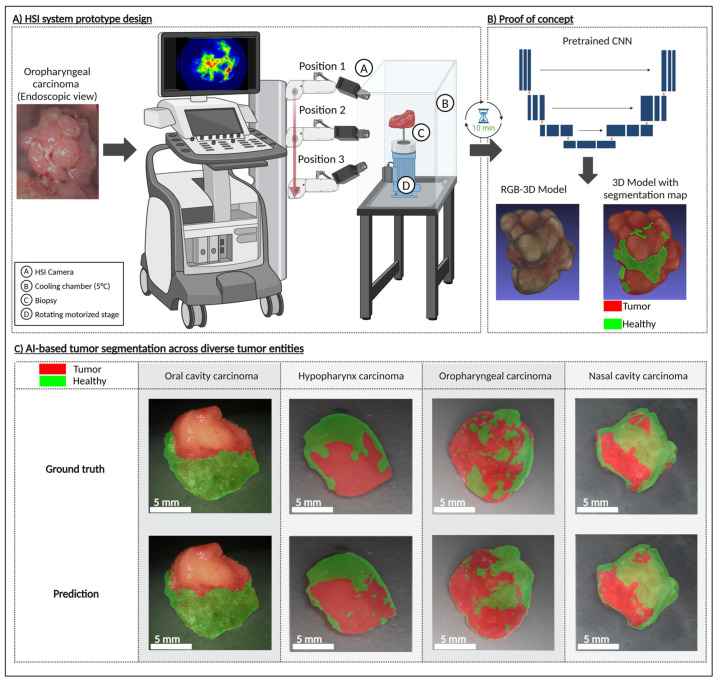
(**A**) Conceptual design of the automated hyperspectral biopsy analysis system consisting of an HSI camera, a cooling chamber, a rotating motorized stage for biopsy positioning, and a deep learning model trained to classify tumor presence based on hyperspectral data. (**B**) Proof of concept for system using an oropharyngeal carcinoma: 3D reconstruction of the RGB model, and segmented tumor regions. (**C**) AI-based segmentation across diverse tumor entities (oral cavity carcinoma, hypopharynx carcinoma, oropharyngeal carcinoma) demonstrated on 3D models, with segmentation performed based on HSI data, thereby validating digital mapping approach in 3D. Created in BioRender Release: 24 April 2025 (BioRender Inc., Toronto, ON, Canada). https://BioRender.com/a01i254.

**Figure 4 cancers-17-01617-f004:**
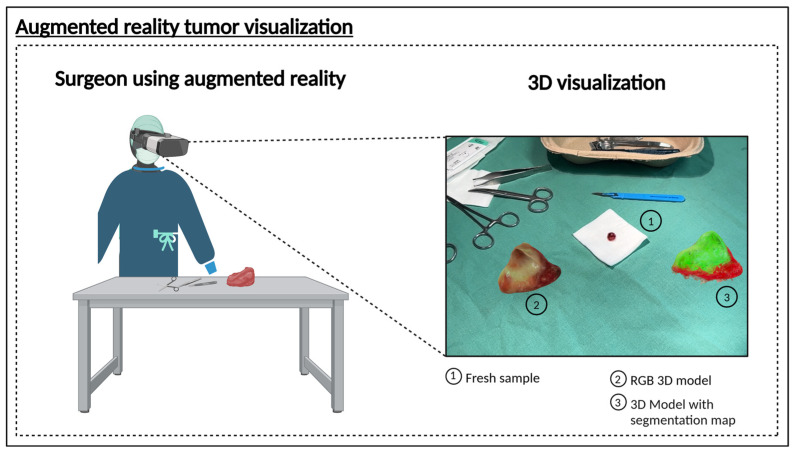
3D model of a HNSCC sample, integrated within an augmented reality environment. Visualization places the model in a spatial context, allowing the surgeon to view the tissue in real time. By virtually rotating the 3D models, the surgeon can examine the sample from various angles, facilitating a detailed analysis of the tissue structure and supporting precise surgical planning. (1) Freshly resected sample. (2) RGB 3D model to visualize the sample. (3) 3D model with the segmentation map based on the HSI information. Created in BioRender Release: 24 April 2025 (BioRender Inc., Toronto, ON, Canada). https://BioRender.com/q09o760.

**Table 1 cancers-17-01617-t001:** Sample and histopathological characteristics of 6 patients. Localization, TNM (tumor, node, metastasis) classification and final histopathological assessment are shown.

ID	Localization	TNM	Histopathology
HSI-3D-1	Oropharynx	pT4a N1 M0	Squamous cell carcinoma
HSI-3D-2	Oral cavity	pT4a N0 M0	Squamous cell carcinoma
HSI-3D-3	Hypopharynx	pT1 N2b M0	Squamous cell carcinoma
HSI-3D-4	Oropharynx	pT4a N3b M0	Squamous cell carcinoma
HSI-3D-5	Nose	pT4a N0 M0	Squamous cell carcinoma
HSI-3D-6	Oral cavity	pT3 N3b M0	Squamous cell carcinoma

**Table 2 cancers-17-01617-t002:** Performance comparison of three model architectures (U-Net, UwU-Net, U-Net-Transformer) based on overall accuracy and class-wise recall, precision, and F1-score.

Model	Class	Accuracy	Recall	Precision	F1-Score
U-Net	Tumor	0.9751 ± 0.0022	0.9325 ± 0.0245	0.9062 ± 0.0106	0.9062 ± 0.0106
	Healthy		0.6857 ± 0.0820	0.7428 ± 0.0347	0.7428 ± 0.0347
	Background		0.9976 ± 0.0005	0.9975 ± 0.0002	0.9975 ± 0.0002
UwU-Net	Tumor	0.9286 ± 0.0165	0.9069 ± 0.1118	0.8390 ± 0.0490	0.8390 ± 0.0490
	Healthy		0.3678 ± 0.2585	0.4161 ± 0.2532	0.4161 ± 0.2532
	Background		0.9921 ± 0.0049	0.9910 ± 0.0026	0.9910 ± 0.0026
U-Net Transformer	Tumor	0.9573 ± 0.0042	0.9141 ± 0.0345	0.8940 ± 0.0116	0.8940 ± 0.0116
	Healthy		0.6570 ± 0.1193	0.6964 ± 0.0573	0.6964 ± 0.0573
	Background		0.9951 ± 0.0012	0.9953 ± 0.0005	0.9953 ± 0.0005

## Data Availability

All presented data will be made available upon reasonable request.

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
