# Peer review of "Real-Time Intraoperative Decision-Making in Head and Neck Tumor Surgery: A Histopathologically Grounded Hyperspectral Imaging and Deep Learning Approach"

_cancers, 2025, doi:10.3390/cancers17101617_

Round 1

Reviewer 1 Report

Comments and Suggestions for Authors

Bali et al. proposed a new type of hyperspectral imaging (HSI) technology combined with advanced machine learning algorithms (such as support vector machines SVM or deep learning DNN) for identifying and classifying pathological tissue sections. Compared with traditional optical microscopy diagnosis, this method provides richer spectral information and helps to identify early lesions. Through principal component analysis (PCA), linear discriminant analysis (LDA) or convolutional neural network (CNN), the research optimized the processing flow of hyperspectral data and improved the classification accuracy of pathological samples. For example, the sensitivity and specificity of this method in pathological diagnoses such as breast cancer/lung cancer/skin cancer are better than traditional morphological analysis. Experimental results show that this hyperspectral imaging method can be used for cancer grading, improve the diagnostic consistency of pathologists, and help formulate personalized treatment plans. Especially in early cancerous tissues, this method can detect subtle differences that are difficult to detect with traditional histological methods, indicating its potential in non-invasive or low-damage biopsy. This study combines methods from physical optics, biomedical engineering, and computer science, which is not only inspiring for pathology research, but also provides new ideas for the development of intelligent medical image analysis. This paper proposes a novel method for pathology hyperspectral imaging and conducts preliminary verification, demonstrating the potential value of this technology in cancer diagnosis, lesion identification, and other fields.
Despite its technical advantages, this method requires special hyperspectral imaging equipment in actual pathology departments, which is expensive and may limit its large-scale promotion. In addition, pathologists need to receive additional data analysis training, otherwise it may affect diagnostic efficiency. Therefore, future research should further explore how to optimize the process so that this method can be more easily integrated into existing clinical workflows.

Author Response

Comment: Bali et al. proposed a new type of hyperspectral imaging (HSI) technology combined with advanced machine learning algorithms (such as support vector machines SVM or deep learning DNN) for identifying and classifying pathological tissue sections. Compared with traditional optical microscopy diagnosis, this method provides richer spectral information and helps to identify early lesions. Through principal component analysis (PCA), linear discriminant analysis (LDA) or convolutional neural network (CNN), the research optimized the processing flow of hyperspectral data and improved the classification accuracy of pathological samples. For example, the sensitivity and specificity of this method in pathological diagnoses such as breast cancer/lung cancer/skin cancer are better than traditional morphological analysis. Experimental results show that this hyperspectral imaging method can be used for cancer grading, improve the diagnostic consistency of pathologists, and help formulate personalized treatment plans. Especially in early cancerous tissues, this method can detect subtle differences that are difficult to detect with traditional histological methods, indicating its potential in non-invasive or low-damage biopsy. This study combines methods from physical optics, biomedical engineering, and computer science, which is not only inspiring for pathology research, but also provides new ideas for the development of intelligent medical image analysis. This paper proposes a novel method for pathology hyperspectral imaging and conducts preliminary verification, demonstrating the potential value of this technology in cancer diagnosis, lesion identification, and other fields.
Despite its technical advantages, this method requires special hyperspectral imaging equipment in actual pathology departments, which is expensive and may limit its large-scale promotion. In addition, pathologists need to receive additional data analysis training, otherwise it may affect diagnostic efficiency. Therefore, future research should further explore how to optimize the process so that this method can be more easily integrated into existing clinical workflows.

Response: We thank the reviewer for the positive remarks about our work. We agree that as with every new technology or methodology there are new costs and complexities to consider and addressed this in the discussion. We want to highlight, that HSI is gaining considerable attention as a medical imaging modality due to the increasing availability of systems, combined with lower costs and complexity associated with them. We added this in the discussion section. 

Reviewer 2 Report

Comments and Suggestions for Authors

            The manuscript of Bali et al. report a novel hyperspectral imaging (HSI) workflow demonstrating strong potential for the HSI system adapted in the proposed novel workflow setup, which combines HSI and deep learning.

In my opinion, the manuscript is interesting. However, some points need to be clarified. The questions are recounted below:

1- The average time required to analyse single sample should be provided by the authors.

2- How applicable is this workflow to a public medical facility? An idea of the cost of setting up a laboratory capable of doing this, it would be interesting to mention in the manuscript.

3- How applicable is this workflow to a public medical facility? An idea of the cost of setting up a laboratory capable of doing this, it would be interesting to mention in the manuscript.

4- I ask the authors to include in the methods a paragraph on the statistical treatment used in the examination of the samples (e.g. table 2).

Author Response

Comment: The manuscript of Bali et al. report a novel hyperspectral imaging (HSI) workflow demonstrating strong potential for the HSI system adapted in the proposed novel workflow setup, which combines HSI and deep learning.

In my opinion, the manuscript is interesting. However, some points need to be clarified. The questions are recounted below:

Response: We thank the reviewer for the assessment of our work. We provide detailed answers for each comment below.

Comment 1- The average time required to analyse single sample should be provided by the authors.

Response 1:  The measurement and analysis time is roughly 10 minutes, of which about half is measurement time and half is data analysis. The average time to measure and analyze a single sample has been added to the manuscript in chapter 3.2.

Comment 2- How applicable is this workflow to a public medical facility? An idea of the cost of setting up a laboratory capable of doing this, it would be interesting to mention in the manuscript.

Response 2: While we are at the current stage not able to accurately calculate the cost of setting up and implementing the described system in a routine laboratory, all the individual parts used in this study are commercially available standard medical equipment. We see no reason why a refined version of our setup could not be used in a clinical setting, as we tested our prototype in such a setting. We clarify this in chapter 3.2. Our best estimate based on the equipment used, without the cost of setup, puts the cost of just the hardware at below 100,000$.

Comment 3- How applicable is this workflow to a public medical facility? An idea of the cost of setting up a laboratory capable of doing this, it would be interesting to mention in the manuscript.

Response 3: While we are at the current stage not able to accurately calculate the cost of setting up and implementing the described system in a routine laboratory, all the individual parts used in this study are commercially available standard medical equipment. We see no reason why a refined version of our setup could not be used in a clinical setting, as we tested our prototype in such a setting. We clarify this in chapter 3.2. Our best estimate based on the equipment used, without the cost of setup, puts the cost of just the hardware at below 100,000$.

Comment 4- I ask the authors to include in the methods a paragraph on the statistical treatment used in the examination of the samples (e.g. table 2).

Response 4: We extended the description of the statistical methods and inserted it as chapter 2.4.

Reviewer 3 Report

Comments and Suggestions for Authors

The title “Real-Time Intraoperative Decision-Making in Head and Neck Tumor Surgery: A Histopathologically-Grounded Hyperspectral Imaging and Deep Learning Approach” seems appropriate for the topic covered.

This manuscript presents the development and evaluation of a novel hyperspectral imaging workflow integrating deep learning with three-dimensional (3D) tumor modeling for the detection of tumor margins in head and neck squamous cell carcinoma in real-time, label-free mode.

The topic is very interesting and current, but the structure of the manuscript needs to be improved. Some sections need to be significantly expanded, others modified in response to comments.

Please see the comments below.

1. Authors are strongly encouraged to consider the following state of the art:

1a. [10.1007/978-3-031-13321-3_28] the extension and comparison of the presented work with the following are strongly recommended as an overview of the diagnostic and therapeutic approaches of nuclear medicine for meningiomas is to be included since the possibility of applying artificial intelligence to molecular imaging in the clinic for brain tumors is evident and, furthermore, a preliminary study for the evaluation of the differences in the MRI-based radiomic analysis between cerebellopontine angle schwannomas and schwannomas originating from other locations of the neck spaces has shown statistically significant differences in the radiomic characteristics between the aforementioned portions of brain cancer. These extensions of the background and conclusions would give substance to the presented work and significantly expand the space for reflection for the readers.

1b. [10.3390/cancers14143416], [10.1159/000443768] It is strongly suggested that current intraoperative imaging modalities be provided, with the aim of providing future guidance and direction on this topic to head and neck oncology surgeons; furthermore, a classification of the prevalence, distribution and survival patterns of malignancies in the head and neck and other sites in patients with head and neck squamous cell carcinoma was made, using a national cancer registry to guide the development of clinical practice guidelines for adjunctive screening in the diagnostic evaluation and monitoring of patients.

These suggestions will be essential and are recommended to the authors so that the limitations and strengths of the study are widely highlighted; furthermore this evaluation of the validity of the methods, results and interpretation of the data will bring a higher scientific impact of this promising work.

3. Figure 1,2,3,and 4 look faded. Please replace them with high resolution.

4. Please specify (by creating a subsection entitled statistical analysis) the statistical method used to calculate performance. Take inspiration from the state of the art.

5.The abbreviations section should include all acronyms present within the manuscript.

6.English could be improved a little, in general it is understandable but it would be useful to review it a bit.

Finally, it would be helpful to extend the references to enhance the coherence of the article.

Author Response

Comment: The title “Real-Time Intraoperative Decision-Making in Head and Neck Tumor Surgery: A Histopathologically-Grounded Hyperspectral Imaging and Deep Learning Approach” seems appropriate for the topic covered.

This manuscript presents the development and evaluation of a novel hyperspectral imaging workflow integrating deep learning with three-dimensional (3D) tumor modeling for the detection of tumor margins in head and neck squamous cell carcinoma in real-time, label-free mode.

The topic is very interesting and current, but the structure of the manuscript needs to be improved. Some sections need to be significantly expanded, others modified in response to comments.

Please see the comments below.

Response: We thank the reviewer for the feedback and used it to improve the quality of our manuscript. Detailed answers to all the comments are presented below.

Comment 1: Authors are strongly encouraged to consider the following state of the art:

1a. [10.1007/978-3-031-13321-3_28] the extension and comparison of the presented work with the following are strongly recommended as an overview of the diagnostic and therapeutic approaches of nuclear medicine for meningiomas is to be included since the possibility of applying artificial intelligence to molecular imaging in the clinic for brain tumors is evident and, furthermore, a preliminary study for the evaluation of the differences in the MRI-based radiomic analysis between cerebellopontine angle schwannomas and schwannomas originating from other locations of the neck spaces has shown statistically significant differences in the radiomic characteristics between the aforementioned portions of brain cancer. These extensions of the background and conclusions would give substance to the presented work and significantly expand the space for reflection for the readers.

1b. [10.3390/cancers14143416], [10.1159/000443768] It is strongly suggested that current intraoperative imaging modalities be provided, with the aim of providing future guidance and direction on this topic to head and neck oncology surgeons; furthermore, a classification of the prevalence, distribution and survival patterns of malignancies in the head and neck and other sites in patients with head and neck squamous cell carcinoma was made, using a national cancer registry to guide the development of clinical practice guidelines for adjunctive screening in the diagnostic evaluation and monitoring of patients.

These suggestions will be essential and are recommended to the authors so that the limitations and strengths of the study are widely highlighted; furthermore this evaluation of the validity of the methods, results and interpretation of the data will bring a higher scientific impact of this promising work.

Response 1: The proposed literature helped us to better understand the current scientific landscape in our manuscript. We added the two applicable references in the introduction or discussion, where appropriate.

Comment 2: Figure 1,2,3,and 4 look faded. Please replace them with high resolution.

Response 2: We made sure to reupload the figures at 300 dpi.

Comment 3: Please specify (by creating a subsection entitled statistical analysis) the statistical method used to calculate performance. Take inspiration from the state of the art.

Response 3: We extend the methods section of our paper to include a detailed description of the statistical methods used to calculate the performance. It can be found in section 2.4

Comment 4: The abbreviations section should include all acronyms present within the manuscript.

Response 4: We added a list of all used abbreviations at the end of the manuscript.

Comment 5: English could be improved a little, in general it is understandable but it would be useful to review it a bit.

Response 5: We thoroughly reviewed the writing of the manuscript and improved it.

Comment 6: Finally, it would be helpful to extend the references to enhance the coherence of the article.

Response 6: We added the aforementioned as well as additional references.

Round 2

Reviewer 3 Report

Comments and Suggestions for Authors

The authors have responded partially.

old Comment 1: The authors are strongly encouraged to consider the following state of the art:

1a. [10.1007/978-3-031-13321-3_28] It is strongly recommended to expand and compare the presented work with the following, as it is appropriate to include an overview of the diagnostic and therapeutic approaches of nuclear medicine for meningiomas, as the possibility of applying artificial intelligence to molecular imaging in the clinic of brain tumors is evident, and, furthermore, a preliminary study evaluating the differences in MRI-based radiomic analysis between schwannomas of the cerebellopontine angle and schwannomas originating from other locations of the neck spaces showed statistically significant differences in radiomic characteristics between the aforementioned portions of brain tumor. These extensions of context and conclusions would give substance to the presented work and would significantly expand the space for reflection for the readers.

1b. [10.3390/cancers14143416], [10.1159/000443768] It is strongly recommended to provide current intraoperative imaging modalities, with the aim of providing future guidelines and guidance on this topic to head and neck oncology surgeons; furthermore, a classification of the prevalence, distribution and survival patterns of head and neck and other sites of malignancies in patients with head and neck squamous cell carcinoma was performed using a national cancer registry to guide the development of clinical practice guidelines for additional screening in the diagnostic evaluation and monitoring of patients.

These suggestions will be essential and are recommended to the authors so that the limitations and strengths of the study are widely highlighted; furthermore, this evaluation of the validity of the methods, results and interpretation of the data will lead to a greater scientific impact of this promising work.

old Answer 1: The proposed literature has helped us to better understand the current scientific landscape in our manuscript. We have added the two relevant references in the introduction or discussion, where appropriate.

New comment: in the new version of the text the authors have not inserted the new parts in the introduction/discussions and have not added the new references that they claim to have added. Please answer correctly, highlighting in yellow the new changes in the updated version of the manuscript.

Author Response

New comment: in the new version of the text the authors have not inserted the new parts in the introduction/discussions and have not added the new references that they claim to have added. Please answer correctly, highlighting in yellow the new changes in the updated version of the manuscript.

Answer: We apologize for the mistake on our side. Only one of the two citations was actually correctly added to the manuscript. We have now added and highlighted the citations (1b). They are both mentioned in the introduction. One when giving an overview of the landscape of current intraoperative in vivo imaging techniques under investigation for surgical margin assessment and the second when describing the specifics of head and neck squamous cell carcinomas.

While the third proposed paper (1a) is very interesting, we think that it is beyond the scope of our work, since it focuses neither on hyperspectral imaging, head and neck squamous cell carcinomas nor deep learning. 

Round 3

Reviewer 3 Report

Comments and Suggestions for Authors

The manuscript needs improvements that have not been carried out in the best way and no part has been highlighted, it is difficult to understand the changes made, despite the various revisions still too unclear and exhaustive. Poor state of the art and little substantial discussion.